# Mediation Effects of Social Cognition on the Relationship between Neurocognition and Social Functioning in Major Depressive Disorder and Schizophrenia Spectrum Disorders

**DOI:** 10.3390/jpm13040683

**Published:** 2023-04-19

**Authors:** Takashi Uchino, Ryo Okubo, Youji Takubo, Akiko Aoki, Izumi Wada, Naoki Hashimoto, Satoru Ikezawa, Takahiro Nemoto

**Affiliations:** 1Department of Neuropsychiatry, Faculty of Medicine, Toho University, 6-11-1 Omori-nishi, Ota-ku, Tokyo 143-8541, Japan; takashi.uchino@med.toho-u.ac.jp (T.U.); youji.takubo@med.toho-u.ac.jp (Y.T.);; 2Department of Psychiatry and Implementation Science, Faculty of Medicine, Toho University, 6-11-1 Omori-nishi, Ota-ku, Tokyo 143-8541, Japan; 3Department of Clinical Epidemiology, Translational Medical Center, National Center of Neurology and Psychiatry, 4-1-1 Ogawa-Higashi, Kodaira, Tokyo 187-8551, Japan; ryo-okubo@ncnp.go.jp; 4Department of Psychiatry, Graduate School of Medicine, Hokkaido University, North 15, West 7, Kita, Sapporo 060-8638, Japan; hashinao@med.hokudai.ac.jp; 5Endowed Institute for Empowering Gifted Minds, Graduate School of Arts and Sciences, University of Tokyo, 3-8-1 Komaba, Meguro-ku, Tokyo 153-0041, Japan; satoru-ikezawa@g.ecc.u-tokyo.ac.jp

**Keywords:** ACSo, cognitive function, depression, PDQ, psychosis, real-world functioning, schizophrenia, SFS, social functioning, subjective cognitive complaints

## Abstract

Background: In schizophrenia spectrum disorders (SSD), social cognition mediates the relationship between neurocognition and social functioning. Although people with major depressive disorder (MDD) also exhibit cognitive impairments, which are often prolonged, little is known about the role of social cognition in MDD. Methods: Using data obtained through an internet survey, 210 patients with SSD or MDD were selected using propensity score matching based on their demographics and illness duration. Social cognition, neurocognition, and social functioning were evaluated using the Self-Assessment of Social Cognition Impairments, Perceived Deficits Questionnaire, and Social Functioning Scale, respectively. The mediation effects of social cognition on the relationship between neurocognition and social functioning were examined in each group. Invariances of the mediation model across the two groups were then analyzed. Results: The SSD and MDD groups had mean ages of 44.49 and 45.35 years, contained 42.0% and 42.8% women, and had mean illness durations of 10.76 and 10.45 years, respectively. In both groups, social cognition had significant mediation effects. Configural, measurement, and structural invariances across the groups were established. Conclusion: The role of social cognition in patients with MDD was similar to that in SSD. Social cognition could be a common endophenotype for various psychiatric disorders.

## 1. Introduction

The social losses associated with mental illness are enormous, and long-term declines in social functioning can impact various social situations, including employment, schooling, and interpersonal relationships [1,2]. Declines in social functioning are relatively common, especially in cases with severe mental illnesses such as psychotic disorders, bipolar disorders, and major depressive disorder, which often have a chronic course [3,4]. In people with schizophrenia, the improvement of social functioning, as well as the improvement of psychiatric symptoms, has long been considered an important therapeutic goal, and factors contributing to social functioning have been explored [5]. Impairments in neurocognition are a core feature of schizophrenia and are representative of contributing factors impacting social functioning [6,7]. The associations between social functioning and various neurocognitive domains, including processing speed, verbal memory, working memory, and divergent thinking, have been clarified, and their biological bases have been widely reported [8,9,10,11]. However, impairments in neurocognition alone cannot fully explain the decline in social functioning, and factors mediating the relationship between neurocognition and social functioning have been explored. One key candidate is social cognition, and many studies have reported that there was a significant indirect relationship between neurocognition and social functioning with social cognition as a mediator [12,13]. Social cognition refers to “the mental operations that underlie social interactions, including perceiving, interpreting, and generating responses to the intentions, dispositions, and behaviors of others” [14,15]. The representative domains of social cognition are the theory of mind, social perception and knowledge, attributional style/bias, and emotion processing [14,16]. Various measures of social cognition have been developed, and a decline in social cognition in people with schizophrenia has been widely reported [17,18,19,20]. The impairment in social cognition in patients with schizophrenia is known to often be associated with a variety of clinical symptoms and could underlie the manifestation of the clinical symptoms in schizophrenia. According to one proposed hypothesis, the negative symptoms in patients with schizophrenia are merely the downstream effects of impaired social cognition on social functioning in real-world settings [21]. Furthermore, impaired social cognition has also been reported to be associated with positive symptoms and other general symptoms. The theory of mind refers to the ability of a person to represent the mental states of others and to interpret their mental states. Impairments of the theory of mind, such as overinterpreting the mental states of others, could cause delusions, although the evidence remains controversial [22]. Attributional bias is defined as the way in which individuals explain the causes and make sense of social events or interactions. Changes in attribution bias are known to be associated with persecutory paranoid symptoms, suspiciousness, as well as social anxiety [23]. Therefore, the impairments in social cognition are attracting attention as a novel therapeutic target for achieving symptomatic and functional recovery [24,25]. It has now been reported that social cognition has not only a simple mediating effect on the relationship between neurocognition and social functioning but also a variety of other factors related to social functioning, and these factors could affect one another. However, neurocognition and social cognition remain critical factors [26,27,28].

In addition to schizophrenia, impairments in cognitive function in other psychiatric disorders have also been observed. In people with major depressive disorder, a wide range of cognitive impairments has been reported [29,30,31]. Importantly, even after the remission of mood symptoms, cognitive impairments persisted and worsened with each recurrent episode [32,33]. These cognitive impairments might be a core feature of major depressive disorder and an independent symptom, rather than a symptom that arises secondarily from mood symptoms [29,34,35]. A growing number of reports on social cognition in major depressive disorder have also reported long-term impairments and associations with social functioning [36,37,38]. As noted above, social cognition in schizophrenia mediates the relationship between neurocognition and social functioning; however, in major depressive disorder, the detailed role of social cognition in social functioning remains poorly understood [39,40]. Establishing strategies to improve social functioning is an important issue not only in schizophrenia but also in major depressive disorder, and there is an urgent need to elucidate mechanisms related to the manifestations of social functioning.

Traditionally, most studies assessing neurocognition and social cognition have used objective measures, in general, neuropsychological tests. The representative neuropsychological tests for measuring neurocognition are the MATRICS Consensus Cognitive Battery (MCCB) [41] and the Brief Assessment of Cognition in Schizophrenia (BACS) [42], which have been reported as being useful to assess a wide range of domains of neurocognition. There are also commonly used tests to measure social cognition, depending on the evaluated domains. For example, the Bell-Lysaker Emotion Recognition Task (BLERT) [43] and Penn Emotion Recognition Task (ER-40) [44] have been used to assess the level of emotional processing, and the Hinting Task [45] and Awareness of Social Inferences Test (TASIT) [46] have been used to assess the level of the theory of mind. On the other hand, there are also reports of studies in which attempts have been made to assess the level of subjective difficulties that the patients themselves experience in association with cognitive impairment. The Observable Social Cognition Rating Scale (OSCARS) has been developed as a self- or informant-reported measure of social cognition [47]. Although the discrepancies in the results between objective and subjective evaluations remain to be explored, the OSCARS is reported as a reliable tool for broadly detecting impaired social cognition. Another questionnaire developed to measure the subjective difficulties associated with impaired social cognition is the Self-Assessment of Social Cognition Impairments (ACSo) [48]. The ACSo has been developed for a wide range of mental illnesses characterized by impaired cognition and has shown to be a useful psychometric tool in studies of patients with schizophrenia, schizoaffective disorder, bipolar disorder, and autism spectrum disorder. A growing number of studies have used ACSo for evaluating patients with psychiatric illnesses [49,50,51]. The Perceived Deficits Questionnaire (PDQ) has been developed as a tool for the subjective assessment of neurocognitive impairment [52]. It has mainly been used in research to evaluate patients with depression, and versions in several languages have been developed and examined for their reliability and validity [53,54]. This questionnaire has also been used in studies to examine the efficacy of antidepressants on cognitive function [55]. Although neuropsychological tests that can objectively measure cognitive function are the gold standard, trained professional staff are needed to administer these tests. In addition, these tests are not always comfortable for the participants, as they could take a long time. Therefore, further study is needed on the usefulness of subjective measures of cognition that have come to be increasingly used in research.

In the present study, we hypothesized that the mediating effects of social cognition on the relationship between neurocognition and social functioning would be similar in both people with schizophrenia and those with major depressive disorder. The purpose of the present study was to examine this relationship in patients with stable schizophrenia spectrum disorders and major depressive disorder in a web-based survey using self-administered questionnaires.

## 2. Methods

### 2.1. Study Design and Participants

This study was part of a previous web-based survey [56]. The cross-sectional survey was conducted from 5 March to 15 March 2021 by a professional agency (Rakuten Insight, Inc., Tokyo, Japan; https://member.insight.rakuten.co.in/ (accessed on 1 March 2023)) and included a large internet survey panel of individuals who had previously enrolled as subjects with various mental and physical illnesses or with no history of illness on a self-reported basis. The registration information for this panel was regularly checked and updated by the agency. The agency sent the survey panel a link to the online question form via an email explaining the survey. The inclusion criteria for the survey participants were an age of 20 to 59 years, a diagnosis of schizophrenia, schizoaffective disorder, or major depressive disorder, a history of continuous outpatient treatment for at least one year, no history of psychiatric hospitalization within three months prior to the study, and being sufficiently stable clinically to answer the questionnaire by themselves. The exclusion criteria were a history of alcohol or substance abuse, brain injury, convulsive seizure, or severe physical illness.

The survey began with the presentation of a page explaining the purpose of the survey and obtaining the participants’ consent, followed by questions regarding the inclusion/exclusion criteria. To confirm the diagnoses, a question was asked regarding whether the participants had been informed by a clinician that they had schizophrenia, schizoaffective disorder, or major depressive disorder; if they had not, they were excluded. Self-reported questions concerning each of the other inclusion/exclusion criteria were also included (i.e., whether the patient had been receiving outpatient treatment for at least one year continuously; whether the patient had been hospitalized within three months; and whether the patient had no episodes of substance abuse, convulsions, or serious physical illness). Participants who met all the criteria were then allowed to proceed to the main question pages; those who did not meet all the criteria exited the survey. After the completion of the survey, respondents who provided the same answer to all the questions (i.e., straight-lining) were excluded. Furthermore, based on a method used for a previous internet survey study [57], participants who provided fraudulent responses were excluded. Specifically, we set up a five-choice question and asked the participants to “please choose the second from the bottom of the following options.” The choices were, in order from top to bottom, A, B, C, D, and E. If the participants chose the wrong answer to this question (i.e., any answer other than D), they were excluded from the analysis. Further details regarding the survey were described in the previous papers [56].

Informed consent was obtained before the participants responded to the questionnaire, and the participants were provided the option to stop the survey at any point. The study protocol was approved by the Ethics Committee of the Faculty of Medicine, Toho University (A20074 and A22041_ A20074). The internet survey agency respected the Act on the Protection of Personal Information in Japan. This study was performed in accordance with the latest version of the Declaration of Helsinki.

Completed survey data were obtained from 232 patients with schizophrenia or schizoaffective disorder and 441 patients with major depressive disorder. Using propensity score matching [58], 210 patients with schizophrenia or schizoaffective disorder (schizophrenia spectrum disorders; SSD group) and 210 patients with major depressive disorder (MDD group) were extracted based on their ages, sexes, and durations of illness. These variables were used in the logistic regression to generate a propensity score. Using the nearest-neighbor method without replacement and a caliper of 0.20, subjects were matched 1:1.

### 2.2. Measures

Social cognition was evaluated using the ACSo. The ACSo is a 12-item self-administered questionnaire examining subjective complaints regarding four domains of social cognitive impairment: emotional processes, social perception and knowledge, theory of mind, and attributional bias [48]. It consists of 12 items on a five-point Likert scale (range of 0 to 48). A higher score on the ACSo indicates a greater difficulty in the social cognition domain. Neurocognition was evaluated using the PDQ [52]. The PDQ is a self-administered questionnaire examining self-perceived neurocognitive difficulties within the domains of prospective memory, retrospective memory, attention/concentration, and planning/organization. It consists of 5 items on a five-point Likert scale (range of 0 to 20). A higher score on the PDQ indicates a greater difficulty in the neurocognition domain. The PDQ-5, which is a shorter version of the PDQ, was used in the presently reported survey. Social functioning was evaluated using the Social Functioning Scale (SFS), which is a self-administered measure of a wide range of community functioning parameters, including withdrawal, interpersonal behavior, pro-social activities, recreation, independence-performance, independence-competence, and employment [59,60,61]. A higher score on the SFS indicates a more favorable functioning. The total SFS score was used in the presently reported study.

### 2.3. Data Analysis

First, the demographics and clinical characteristics were compared between the SSD and MDD groups using the *t*-test for continuous variables and the chi-squared test for categorical variables. Then, the mediation effects of social cognition on the relationship between neurocognition and social functioning were examined in each group using Structural Equation Modeling (SEM). To assess the model fit, the chi-squared/df, the comparative fit index (CFI), and the root mean square error of approximation (RMSEA) were calculated. A good fit was regarded as a chi-squared/df value < 2, a CFI > 0.97, and an RMSEA < 0.05. An acceptable fit was regarded as a chi-squared/df < 3, a CFI > 0.95, and an RMSEA < 0.08 [62,63]. In SEM, the model-building approach was used when the models were improved by adding passes while considering the modification indices [64]. The model was adopted if the chi-squared value was significantly (*p* < 0.05) improved by adding a pass. The indirect effect of neurocognition on the social functioning of each group was also assessed using the Sobel test. Finally, invariances of the mediation model across the two groups were examined using a multiple-group SEM. Configural, metric, scalar, residual, and structural invariances across the groups were confirmed step-by-step. To assess the deterioration of the model fit between the configural, metric, scalar, residual, and structural models, changes in CFI (ΔCFI) of <0.01 and changes in RMSEA (ΔRMSEA) of <0.015 were regarded as acceptable [65]. The significance of differences in the chi-square values depends on the size of the sample. The larger the sample, the more likely it is for the difference in the chi-square values between models to be skewed toward significance [66]. Therefore, the invariances of the models could be rejected excessively strongly. Since the total number of subjects in the present study was 420, which may not be considered as being too small, we decided to examine the invariances using several other goodness-of-fit indices (i.e., CFI and RMSEA) that are not affected by the sample size.

Statistical differences were determined using two-tailed tests and a significance level of *p* < 0.05. Data were analyzed using SPSS, version 26.0, and AMOS, version 26.0.

## 3. Results

### 3.1. Demographics and Clinical Characteristics

The demographics and clinical characteristics are shown in Table 1. The SSD and MDD groups had mean (standard deviation) ages of 44.49 (8.27) and 45.35 (9.10) years, were comprised of 42.0% and 42.8% females, and had mean (standard deviation) illness durations of 10.76 (8.90) and 10.45 (8.53) years, respectively. The SSD group had significantly greater difficulties than the MDD group for the PDQ-5 Prospective memory, ACSo Emotional processes, ACSo Theory of mind, and ACSo Social perception scales.

### 3.2. Mediation Effects of Social Cognition

In each group, social cognition had significant mediation effects. In the SEM model-building process, we added the covariance between the Retrospective memory and Planning/Organization 2 to their respective models, taking into account the modification indices, and found that the chi-squared values were significantly improved in both models. The final models for the SSD and MDD groups showed good and acceptable fit indices: respective chi-squared/df values of 1.960 and 2.378, CFI values of 0.981 and 0.974, and RMSEA values of 0.068 and 0.081, respectively (Figure 1). The Sobel tests revealed significant indirect effects of neurocognition on social functioning in both groups (Sobel test statistic = −3.407, *p* = 0.001 in the SSD group; Sobel test statistic = −2.718, *p* = 0.007 in the MDD group).

### 3.3. Invariances in the Mediation Models across Groups

Configural invariance was first examined by specifying the same mediation model across the SSD and MDD groups, while allowing all other parameters to differ. The fit indices for this model were all acceptable (CFI = 0.978, RMSEA = 0.052), suggesting that a configural invariance of the mediation model across the two groups was established.

Then, metric invariance was examined by requiring the same model and equal factor loadings across the two groups, while all other parameters were allowed to differ. The fit indices were all acceptable, and the CFI and RMSEA were not worsened (∆CFI < 0.001, ∆RMSEA = −0.003), suggesting that metric invariance was established.

Scalar invariance was next examined by requiring the same model and equal factor loadings and item intercepts across the two groups, while all other parameters were allowed to differ. The fit indices were all acceptable, and the deteriorations in CFI and RMSEA were within acceptable ranges (∆CFI = −0.003, ∆RMSEA < −0.001), suggesting that scalar invariance was established.

Next, residual invariance was examined by requiring the same model and equal factor loadings, item intercepts, and error variances across the two groups, while all other parameters were allowed to differ. The fit indices were all good, and the deteriorations in CFI and RMSEA were within acceptable ranges (∆CFI = −0.002, ∆RMSEA = −0.001), suggesting that residual invariance was established.

Finally, structural invariance was examined by requiring the same model and equal factor loadings, item intercepts, error variances, and path coefficients across the two groups. The fit indices were all good, and the deteriorations in CFI and RMSEA were within acceptable ranges (∆CFI = 0.001, ∆RMSEA = −0.002), suggesting that structural invariance was established.

After examining the invariances of the mediation model across the two groups, all the models at each level of parameter constraint were accepted, and robust invariances were established. The results of the invariance examinations are shown in Table 2.

## 4. Discussion

This study examined the mediation effects of social cognition in patients with schizophrenia spectrum disorders or major depressive disorder. When the two groups were compared, the SSD group showed greater difficulties than the MDD group for some neurocognition scales and for many social cognition scales. Although mixed results have been reported in comparing cognitive function between schizophrenia and major depressive disorder, in general, the levels of cognitive impairments are more severe in patients with schizophrenia than in patients with major depressive disorders [67,68,69]. Furthermore, social cognition is more impaired than neurocognition in patients with schizophrenia but not in patients with affective disorders [70], which is consistent with the results of the present study. On the other hand, a growing number of studies have sought to clarify cognitive impairment profiles in patients with schizophrenia and major depressive disorder in detail, and further investigation is expected [71,72,73].

Mediation effects of social cognition on the relationship between neurocognition and social functioning have been reported in patients with schizophrenia. The mediation model used in the present study is typical for schizophrenia research. The significant mediation effect of social cognition in the present study is consistent with the robust results of previous studies in patients with schizophrenia [13,26]. Notably, the present study also showed significant mediation effects in patients with major depressive disorder. To the best of our knowledge, no study has demonstrated a mediation effect of social cognition in major depressive disorder that is similar to the mediation effect seen for schizophrenia spectrum disorders. Both schizophrenia and major depressive disorder are characterized by impairments of neurocognition and social cognition, and it is also common for patients with schizophrenia to present with depressive symptoms [74,75]. In addition to the commonality of the symptoms, the present results suggested that social cognition has a similar impact on social functioning in both schizophrenia spectrum disorders and major depressive disorder.

Furthermore, in the examinations of the invariances of the mediation model across the SSD and MDD groups, all the models at each level of parameter constraint were accepted, and robust invariances were established. In recent years, classification issues arising from conventional diagnostic systems based mainly on symptomatology have been pointed out, and there have been attempts to explore biological factors across illnesses [76,77]. Not only do schizophrenia and major depressive disorder share common symptoms, but they also reportedly share a common genetic background [78,79,80]. Therefore, it is not surprising that cognitive impairments are a common endophenotype in both illnesses, and the results of the present study lend support to this conclusion. A previous study reported similar cognitive abnormalities in relation to information processing speed and similar reductions in the gray matter volume in the right medial superior frontal cortex associated with executive functions between patients with schizophrenia and major depressive disorder [81]. These results could be consistent with our finding of the absence of any difference in the subjective difficulty with Planning/Organization as assessed by the PDQ, which is thought to reflect executive functions, between patients with schizophrenia spectrum disorders and major depressive disorder in the present study. This may be the common pathophysiological basis for both disorders, and the subjective difficulties were also consistent. Furthermore, there were no differences in the subjective difficulties related to attribution bias between patients with schizophrenia spectrum disorders and major depressive disorder in this study. As described in the Introduction section, attributional bias is associated not only with delusions, one of the characteristic symptoms of psychosis, but also with social anxiety [23]. In contrast to the traditional classification of neurosis and psychosis, some reports suggest a closer association between social anxiety and persecutory delusions [82]. The present results indicate that schizophrenia spectrum disorders and major depressive disorder share common difficulties with attributional bias and that this could underlie the characteristic impairments in cognition in these patients. This commonality could be applicable not only to major depressive disorder and schizophrenia spectrum disorders but also to various other mental illnesses. Conversely, social cognitive impairments in many other domains (i.e., theory of mind, emotion processes, and social perception) may be specific indicators of schizophrenia spectrum disorders.

The present study had some limitations. The study was conducted as an online survey; consequently, the subjects’ diagnoses were based only on self-reported information. Although this study took a cautious view of adopting an online method and implemented methodologies that enhance the reliability and validity of such surveys [56,57], these precautions may have been insufficient. Additionally, psychiatric symptoms, including depressive mood, were not assessed or included in the model. Mood symptoms are an important contributor to social functioning and social cognition in patients with schizophrenia spectrum disorders and major depressive disorder. For example, it has been reported that a higher level of depressive symptoms predicted lower social functioning and higher social cognitive difficulties in patients with schizophrenia [83]. However, the fact that we were able to confirm the same mediation model for the self-reported schizophrenia spectrum disorder patients in the present study as that reported in previous studies may indicate a certain degree of validity for the present participants. Regarding the measures used in the present study, all the items were self-administered and examined subjective difficulties. It is not uncommon for the results of an evaluation of cognitive impairment to differ between objective assessments and subjective ratings. A previous study reported that the greater the degree of cognitive impairment in patients with schizophrenia or schizoaffective disorder, the more likely they were to overestimate their level of functioning. On the other hand, the greater the severity of symptoms in persons with depression, the more likely they were to underestimate their level of functioning. The level of depressive mood could be associated with the introspective accuracy of a patient with depression [84]. However, there are also paradoxical reports that a better insight might be associated with higher severity of depressive symptoms [85]. It appears that the relationship between the severity of depressive symptoms and introspective accuracy may not be unidirectional but rather more complex. The ACSo, which is a measure of subjective social cognitive difficulties, had a high degree of relationship with social functioning and may help to examine the role of social cognition on social functioning [56]. Considering the differences in the results between subjective and objective assessments and the confounding factors, it is important to interpret the results of the present study with caution, and further research conducted using objective assessment measures is warranted.

## 5. Conclusions

The role of social cognition in major depressive disorder was similar to that observed in schizophrenia. In both schizophrenia and major depressive disorder, the mediation effect of social cognition on the relationship between neurocognition and social functioning was confirmed to have robust invariances. Social cognitive impairments could be a common endophenotype for various mental illnesses.

## Figures and Tables

**Figure 1 jpm-13-00683-f001:**
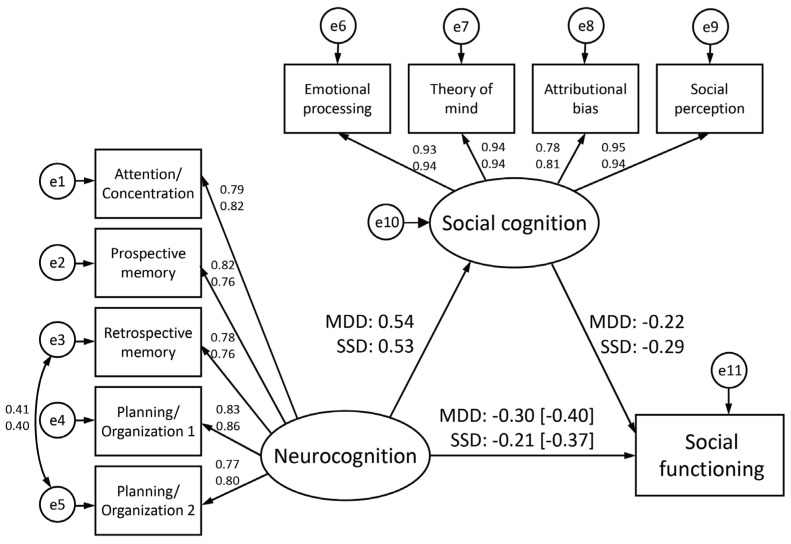
Mediation effects of social cognition on the relationship between neurocognition and social functioning in patients with schizophrenia spectrum disorders or major depressive disorder. The numbers in square brackets are standardized coefficients of the direct effect of neurocognition on social functioning. The numbers in the upper row are the standardized coefficients for the major depressive disorder group, while those in the lower row are the standardized coefficients for the schizophrenia spectrum disorders group.

**Table 1 jpm-13-00683-t001:** Demographics and clinical characteristics of patients with schizophrenia spectrum disorders or major depressive disorder.

	SSD (*N* = 210) Mean (SD) or %	MDD (*N* = 210) Mean (SD) or %	*p* Value
Age	44.49 (8.27)	45.35 (9.10)	0.31
Sex (male/female)	58.0/42.0	57.2/42.8	0.84
Duration of illness	10.76 (8.90)	10.45 (8.53)	0.71
PDQ-5 Attention/concentration	2.47 (1.34)	2.51 (1.27)	0.74
PDQ-5 Prospective memory	2.37 (1.25)	2.10 (1.23)	0.03 *
PDQ-5 Retrospective memory	1.98 (1.18)	1.81 (1.15)	0.14
PDQ-5 Planning/organization 1	2.39 (1.25)	2.40 (1.23)	0.91
PDQ-5 Planning/organization 2	2.03 (1.21)	1.87 (1.16)	0.15
ACSo Emotional processes	3.67 (3.05)	3.10 (2.73)	0.05 *
ACSo Theory of mind	4.45 (3.15)	3.86 (3.00)	0.05 *
ACSo Attributional bias	4.07 (3.32)	3.70 (2.87)	0.23
ACSo Social perception	3.77 (3.02)	3.18 (2.79)	0.04 *
SFS total score	102.31 (25.14)	106.90 (24.22)	0.06

* *p* < 0.05; ACSo, Self-Assessment of Social Cognition Impairments; MDD, major depressive disorder; PDQ-5, Perceived Deficits Questionnaire 5; SFS, Social Functioning Scale; SSD, schizophrenia spectrum disorders.

**Table 2 jpm-13-00683-t002:** Invariances of the mediation models across groups using Multi-group Structural Equation Modeling.

	*X2* (*df*)	*X2*/*df*	Δ*X2* (*df*)	CFI	ΔCFI	RMSEA	ΔRMSEA	Judgment
Configural invariance	138.816 (65)	2.136	Ref	0.978	Ref	0.052	Ref	ACCEPT
Metric invariance	144.458 (72)	2.006	5.642 (7)	0.978	<0.001	0.049	−0.003	ACCEPT
Scalar invariance	164.690 (82)	2.008	20.232 (10) *	0.975	−0.003	0.049	<0.001	ACCEPT
Residual invariance	184.019 (93)	1.979	19.329 (11)	0.973	−0.002	0.048	−0.001	ACCEPT
Structural invariance	184.615 (97)	1.903	0.596 (4)	0.974	0.001	0.046	−0.002	ACCEPT

* *p* < 0.05; CFI, Comparative Fit Index; RMSEA, Root Mean Square Error of Approximation.

## Data Availability

Data supporting the findings of this study are available upon reasonable request to the corresponding author. The data will not be made publicly available because of privacy and ethical restrictions.

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
