# Peer review of "Mediation Effects of Social Cognition on the Relationship between Neurocognition and Social Functioning in Major Depressive Disorder and Schizophrenia Spectrum Disorders"

_jpm, 2023, doi:10.3390/jpm13040683_

Round 1

Reviewer 1 Report

This is an interesting study. The main concerns that I had with the study design are commented upon by the authors in their discussion. They appear well aware of the limitations of this design. Nevertheless the results are interesting and should motivate other research groups to carry out such studies using improved designs.  I think it will be important to assess these various functions using more methods than just self reports - the presence of depression could bias the self reports in a negative direction - more objective assessments of cognition and social cognition would make for a more convincing case.  

Reviewer 2 Report

The manuscript reports findings from a cross-sectional investigation of the mediating role of social cognition in the relationship between neurocognition and social functioning, which is a timely, trans-diagnostic investigation of the relationship in schizophrenia and major depressive disorder. Individuals with schizophrenia (and schizoaffective disorder) and major depressive disorder were recruited via an online survey, in which participants completed self-report measures of these variables. Both groups of patients were matched with propensity score matching. The analyses were clearly described and well-executed. The authors offered evidence supporting the hypothesized mediating role of social cognition, which was found to be present in both groups of patients. 

My main concern is the use of online survey has limited data collection to self-report measures, which do not necessarily corresponds to objective, “golden-standard” measures of neurocognition and social cognition assessing with behavioural paradigms (though the authors had addressed this in the Discussion). As the authors mentioned in the Introduction, the relationship between neurocognition, social cognition and functioning has been well addressed in schizophrenia, using objective measures of neurocognition and social cognition, which do not tap into the same constructs as these “presumed” self-report measures. The authors could cite evidence of the validity of these measures in individuals with schizophrenia and major depressive disorders, which may increase the credibility and value of the findings. I understand these the design did not allow the use of more optimal measures of neurocognition and social cognition, but these issues should be stated more explicitly for the readers’ consideration in the Introduction and Disucssion.

My other minor comments are as follows:

Introduction:

-        Due to the extensive discussion of the role of social cognition in schizophrenia symptoms (for example theory of mind deficits and persecutory delusions), the authors could mention some of these lines of research to highlight the relevance of social cognition deficits in schizophrenia.

-        The phrase “chronic-phase” was used throughout the manuscript. How was the chronicity of the illness determined? For example, were the individuals in the schizophrenia group NOT in their first episode?

Methods:

-        The authors also include individuals with schizoaffective disorder as an inclusion criterion of the “schizophrenia” group lines 119 – 120, p. 3), which should not cover schizoaffective disorder. The authors may need more careful about the use of the label(s) of diagnosis(es) throughout the whole manuscript, as the title suggests “schizophrenia” only. Maybe “schizophrenia spectrum disorders” is a more accurate description of the sample. Or the authors may only include individuals with schizophrenia only in the analyses.

Results:

-        I noticed a significant chi-square likelihood ratio test for the test of scalar invariance. The authors should justify why the chi-square likelihood ratio test was not used for the determination of measurement invariance.

Discussion:

In reference to the statement “it is not surprising that cognitive impairments are a common endophenotype for both illnesses” (lines 252-253, p. 6), the Discussion may benefit from some elaboration on the common and distinct neurocognition and social cognition underlying schizophrenia and major depressive disorder. How do the current findings suggest deficits in neurocognition, as well as social cognition, as common features or even psychopathological pathways between these two disorders? Distinct patterns of these deficits may underline these two disorders (especially in the literature of the distinction between neurosis and psychosis), which warrant further discussion here.

Round 2

Reviewer 2 Report

Thank you so much for the revision and responses to my comments. As noted in my initial review, the manuscript reports a cross-sectional investigation of the mediating role of social cognition in the relationship between neurocognition and social functioning in individuals with schizophrenia spectrum disorders and major depressive disorder. The manuscript addresses an important and highly clinically relevant area of study, with potential advancement in understanding the role of deficits in neurocognition and social cognition in functioning across mental disorders. The authors are commended for providing a thorough response to my suggestions and a clear summary of the revisions. I have no further recommendations.